# Interdependence between Nuclear Pore Gatekeepers and Genome Caretakers: Cues from Genome Instability Syndromes

**DOI:** 10.3390/ijms25179387

**Published:** 2024-08-29

**Authors:** Lidia Larizza, Elisa Adele Colombo

**Affiliations:** 1Experimental Research Laboratory of Medical Cytogenetics and Molecular Genetics, IRCCS Istituto Auxologico Italiano, Via Ariosto 13, 20145 Milan, Italy; 2Genetica Medica, Dipartimento di Scienze Della Salute, Università Degli Studi di Milano, 20142 Milano, Italy; elisaadele.colombo@unimi.it

**Keywords:** *NUP98* nucleoporopathy, nucleoporins, NPC damage, Rothmund–Thomson syndrome, Werner syndrome, POIKTMP, genome instability syndromes, gatekeepers, caretakers

## Abstract

This review starts off with the first germline homozygous variants of the Nucleoporin 98 gene (*NUP98*) in siblings whose clinical presentation recalls Rothmund–Thomson (RTS) and Werner (WS) syndromes. The progeroid phenotype caused by a gene associated with haematological malignancies and neurodegenerative disorders primed the search for interplay between caretakers involved in genome instability syndromes and Nuclear Pore Complex (NPC) components. In the context of basic information on NPC architecture and functions, we discuss the studies on the interdependence of caretakers and gatekeepers in WS and Hereditary Fibrosing Poikiloderma (POIKTMP), both entering in differential diagnosis with RTS. In WS, the WRN/WRNIP complex interacts with nucleoporins of the Y-complex and NDC1 altering NPC architecture. In POIKTMP, the mutated FAM111B, recruited by the Y-complex’s SEC13 and NUP96, interacts with several Nups safeguarding NPC structure. The linkage of both defective caretakers to the NPC highlights the attempt to activate a repair hub at the nuclear periphery to restore the DNA damage. The two separate WS and POIKTMP syndromes are drawn close by the interaction of their damage sensors with the NPC and by the shared hallmark of short fragile telomeres disclosing a major role of both caretakers in telomere maintenance.

## 1. Scope of the Review

This review, primed by the finding that the germline mutations of the Nuclear Pore Complex (NPC) nucleoporin *NUP98* gene (MIM*601021) [1] phenocopy the progeroid Rothmund-Thomson (RTS, MIM#618625 [1] for type 1 and MIM#268400 [1] for type 2) genome instability syndrome [2], will investigate the NPC components reported to be involved in direct or indirect interactions with functional and dysfunctional genome caretakers. It will focus on the WRN (RecQ Protein-Like 2, RECQL2; MIM*604611 [1]) helicase and the FAM111B (Family with Sequence Similarity 111, Member B; MIM*615584 [1]) serine protease, responsible—when mutated—for Werner (WS; MIM#277700 [1]) and Hereditary Fibrosing Poikiloderma (POIKTMP; MIM#615704 [1]) syndromes, both exhibiting a clinical overlap with RTS and NUP98 nucleoporopathy. The NPC distribution and the network of NPC components binding the WRN/WRNIP (WRN Interacting Protein) complex [3] and FAM111B [4], either directly or through the NUP93 and NUP98 linkers, will be showcased along with the hallmarks of genomic instability shared by these two apparently discrete syndromes. The interdependence of NPC gatekeepers and genome caretakers, initially reported in yeast [5] and later emerging in human DNA repair diseases, raises outstanding issues on the multifaceted aspects of DNA damage response whose impairment promotes disease and cancer.

## 2. Architecture of the Nuclear Pore Complex

The nuclear envelope (NE) is a hallmark feature of the eukaryotic cell, providing the protective barrier and mechano-transduction interface between the nucleus and the cytoplasm [6,7]. Besides separating genomic DNA from the rest of the cell and transcription from translation, the NE also acts as a hub at the nuclear periphery for nucleocytoplasmic transport (NCT) [8], the regulation of the cell cycle [9], gene expression [10], epigenetic regulation [11], DNA repair, and genome maintenance [12,13,14].

The NE is composed of the nuclear lamina, a fibrillary meshwork made up of type A and B lamins [15], and a double-layered membrane, the inner and outer nuclear membranes, merging at the sites of Nuclear Pore Complexes, which serve as the sole gateways for the transport of macromolecules out of and into the nucleus [8]. The human NPCs, which form around 2000 channels at the NE in one human cell, are supramolecular structures of 120 megadaltons, composed of multiple copies (from 8 to 64) of 34 different proteins termed nucleoporins, followed by a number reflecting the molecular mass, which assemble into sub-complexes with a concentric cylindrical organization and eight-fold symmetry along the nucleocytoplasmic axis, yielding a copy number of eight or a multiple of eight for each of the various nucleoporins. In recent years, the molecular architecture of the human NPC proteinaceous machine has been revealed at increasing resolutions [6,16,17,18,19,20]. Schematically, the NPC is composed of a symmetrical central core and the peripheral asymmetric appendages that make up the cytoplasmic filaments (CF) and the nuclear basket (NB) (Figure 1a [21]). The central core comprises four concentric rings: an inner ring (IR), interwoven by linker interactions, which encircles the central channel, two outer nuclear and cytoplasmic rings (NR and CR) formed by the nuclear and cytoplasmic coat nucleoporin complex (CNC or Y-complex or Nup107–160 complex), and the luminal ring surrounding the NPC where Pore Membrane (POM) proteins anchor the outer rings to both sides of the nuclear envelope. According to their location within the NPC, Nups are classified as transmembranes or pore membranes of the luminal ring, scaffold of the IR, NR, and CR, and FG (Phenylalanine-Glycine)-rich nucleoporin components of NB, CF, IR, and the central pore channel. The FG-Nups account for almost one-third of Nups and contain large intrinsically disordered regions (IDRs) rich in hydrophobic FG repeats linked by uncharged spacers [22], which form the hydrogel-like barrier of the central channel (Figure 1a) [8].

A near-atomic resolution has been achieved for most of the evolutionarily conserved NPC linker scaffold of the central core by the powerful combination of cryo-electron microscopy with machine learning prediction and biochemical reconstitution [19,23,24]. These advances have allowed researchers to define the nucleoporin complexes of the central core rings, CF and NB of human NPC (Figure 1b), and the subcomponents and their interactions in the scaffold core and CF (Figure 2a). The composition of the best-studied Y-shaped coat nucleoporin complex of cytoplasmic and nuclear outer rings is shown in Figure 1c.

## 3. Functions of the Nuclear Pore Complex

NCT through NPCs is the primary function, and it is fundamental to all other NPC functions. Essential to NCT are the unstructured FG-rich Nups, which are projected from the inner ring into the central channel and form a hydrogel [25,26] acting as a sieve-like permeability barrier within the NPC (Figure 1a). Soluble small molecules can pass through by means of passive diffusion while the “selective” passage of molecules >40 KDa requires binding to a range of nuclear transport receptors (NTRs), mainly of the karyopherins family (importins, exportins, biportins), whose affinity for FG repeats and the ultrafast exchange kinetics allow the bound cargo to traverse the diffusion barrier [8,20,27,28].

NUP98, with its N-ter portion, is involved in hydrogel constitution and may facilitate the translocation of the transport complex across the permeability barrier due to the high density of its most cohesive FG domains, the additional binding sites for NTRs [29], and the highly conserved Gle2-binding sequence (GLEBS) motif that binds RAE1 (Ribonucleic acid export 1), a component of the RNA export machinery [30]. NUP98, the unstructured NUP35 (alias NUP53), and other members of the NUP93 complex (Figure 1b) act as linkers holding together the inner ring and connecting it to the scaffold Nups which are concentrically layered around it [20,31]. This architectural framework is able to accommodate the structural changes due to reversible dilation–contraction of the central channel, which result from the tension imparted to the nuclear envelope by mechanical stress/hypertonic shock [32]. The flexibility of modulating the diameter (~200 Å) of the central channel, with the consequent varying levels of transportability, highlights an emerging role of the NPC in mechano-transduction [33].

The NCT is dependent on several other macromolecules and proteins, and one such crucial protein is the Ras-related nuclear protein RAN, a small GTP-ase that regulates the speed and direction of transport by setting the intracellular GTP gradient that decreases ~ 200-fold from nucleus to cytoplasm [34].

As regards trafficking into and out of the nucleus of large macromolecular complexes, such as the core DNA replication machinery, RNA polymerase II, and the ribosomes, our knowledge is limited by a poor understanding of how their assembly/disassembly is coordinated with their function [28].

The disordered transport machinery has been visualized in the nanosized NPC central channel by probing conformations of FG-NUP98 in live cells and permeabilized cells and coupling molecular modeling to fluorescence microscopy [35].

Beyond the well-documented role of NPC in NCT and NPC disassembly/reassembly during mitosis and cell cycle [8], growing evidence points to “off-pore” functions of Nups of vital importance for cell life. The active role of NPC in organizing open chromatin has been disclosed by much evidence of various cell-type specific chromatin binding patterns of NPC components [10,36,37]. Physical interactions of Nups with chromosomal loci have been shown to affect the 3D arrangement of genes and the folding of topologically associated domains, pointing to the NPC as a regulator of genome architecture and a platform for transcription factors and epigenetic modulators [11].

Nups are integral players in the nucleoplasm, acting in the vicinity of the NPC as DNA repair hubs where hard-to-repair double-strand breaks (DSBs) or intermediates are relocated and specific repair pathways are exploited to restore genome integrity [13,14,38]. Studies of DSB repair in *S. Cerevisiae* cells at advanced replicative ages have revealed that repair by homologous recombination (HR) is the first to decrease with aging and is replaced by non-homologous end-joining (NHEJ). When this error-prone pathway is also not functioning, the break sites associate with the NPC at the nuclear periphery, yielding a high frequency of repair products with genomic mutations [39]. The accumulation of damage at NPCs, nuclear leakiness, and increased genomic instability have been identified as the main nuclear changes contributing to aging and premature aging [7,40]. In keeping with these findings, the majority of diseases with accelerated or premature aging are caused by defective DNA repair [41,42,43].

## 4. The NPC and Human Disease

The disruption of the NPC or components of the nuclear-cytoplasmic transport machinery has been recognized as a major cause of human disease, both inherited and acquired.

Germline pathogenic variants in components of the NPC have been increasingly implicated in a wide range of rare developmental disorders, revealing proneness to mutation of scaffold-linker and cytoplasmic filaments Nups [44].

Well-studied are the recurrent chromosomal translocations of *NUP98* and *NUP214* genes that are often rearranged in leukemia [45,46]. The oncogenic properties of the new chimeric proteins, containing the IDR region of the FG-Nups fused to homeodomains or domains with a role in transcriptional or epigenetic regulation of several partners [47], have been related to the ability of the invariably retained FG repeats to self-assemble into phase-separate nuclear bodies [48] and condensate interacting molecules, altering the 3D genome structure at target sites driving leukemogenesis [49,50].

Furthermore, impaired nucleocytoplasmic transport has emerged as a prominent mechanism of several neurodegenerative disorders and Nuclear Pore Complexes have been indicated as “a doorway” to neural injury leading to “aging” brain disorders [8,51]. In Alzheimer’s Disease, a direct interaction of the misfolded and aggregated tau protein with FG-NUP98 has been highlighted, and targeting this interaction raises potential therapeutic perspectives [52,53].

NCT and NPC integrity have been found to be compromised in Amyotrophic Lateral Sclerosis (ALS) and other neurodegenerative disorders by the dysregulation of the C-ter structured domain of NUP62. In ALS, caused by *FUS* (Fused in Sarcoma) mutations, the enrichment of FUS-NUP62 C-ter interactions in the cytoplasm has been reported to promote their co-phase separation into amorphous assemblies [54], thus interfering with the localization of NUP62 to the NPC, resulting in NTC defects. Alterations of NPC components triggered by the impaired export and nuclear accumulation of CHMP7, a critical mediator of NPC, have been shown in Induced Pluripotent Stem Cell-derived ALS motoneurons, to initiate the pathological cascade leading to the mis-localization of TAR DNA binding protein 43 (TDP-43), the hallmark of genetic and sporadic forms of ALS and related neurodegenerative disorders [55].

## 5. Bridging Genome Instability Syndromes to the Nuclear Pore

The scope of this review is to link literature reports on the interdependence of Nups gatekeepers and genome caretakers, whose loss or altered function underpins genome instability syndromes. As mentioned above, this issue was triggered by the identification of the first “germline” biallelic variants of *NUP98* affecting the N-ter FG-domain in two adult siblings manifesting signs of aging in early life [2]. The probands’ skin, hair, teeth, and bone defects notably overlapped with the hallmark signs of the autosomal recessive genome instability Rothmund–Thomson syndrome type 1 (RTS1) and type 2 (RTS2) [56]. A major resemblance to RTS1 was scored by the siblings’ distinctive RTS1 features, in particular early-onset bilateral cataracts and lack of osteosarcoma/skin cancer frequently developed by RTS2 patients. The clinical overlap of *NUP98* nucleoporopathy and RTS1 is unsurprising given the complementary function of the proteins coding by the *NUP98* gene and the *ANAPC1* gene (MIM*608473) [1], mutated in RTS1: *ANAPC1* encodes the C1 scaffold subunit of the Anaphase-Promoting Complex or Cyclosome APC/C ubiquitin-ligase [57], whose timely entry in mitosis is regulated by the NUP98-RAE1 complex [58]. RTS2 is caused by biallelic mutations of the *RECQL4* (RECQ Like Helicase 4, MIM*603780) [1] gene encoding a member of the evolutionarily conserved RecQ family of 3′ to 5′ DNA helicases, which perform a critical role in the maintenance of genome stability, the prevention of senescence, and cancer [59]. The progeroid signs shared by *NUP98*-mutated sibs and RTS1/RTS2 individuals are also the hallmarks of Werner syndrome, which is characterized by accelerated aging, along with an increased incidence and early onset of sarcomas. WS is caused by loss of function mutations in the *WRN* gene coding for the WRN/RECQL2 helicase [60], which is identified to possess functional similarities with RECQL4 in modulating DSB repair pathway choices in a cell cycle-dependent way [61,62]. The overlapping clinical spectra of RTS1/RTS2/WS and *NUP98* nucleoporopathy reflect the merging pathways of the underlying genes, whose mutations lead to the faulty repair of DNA lesions, DNA damage accumulation, and hypersensitivity to DNA damaging agents. The findings in yeast, that mutations in several Nups cause sensitivity to DNA damage [63], and the ground-breaking discovery that NPCs can form at nuclear periphery hubs for molecular transactions at damaged DNA sites [5] lead to the notion that *NUP98* nucleoporopathy might be included in the ranks of genome instability syndromes. In anticipation of further cases of this new syndrome, this review will investigate the connections between gatekeepers and caretakers reported in genome instability syndromes clinically resembling *NUP98* nucleoporopathy, to support the paradigm of mutated NUP98 N-ter FG domain in “inherited” disease.

To highlight the interactions between proteins responsible for the maintenance of genome stability and nucleoporins, we will illustrate the link in WS between WRN/WRNIP and NPC, the nucleoporins involved, and the effect of their interactions on NPC structure and nuclear compartments [3,64].

Then we will discuss the link between NPC and FAM111B, which is responsible, when mutated, for Hereditary Fibrosing Poikiloderma [65,66], an autosomal dominant multisystem disorder exhibiting a partial clinical overlap with RTS [56], which shares with WS the hallmark of short fragile telomeres [4].

## 6. *NUP98* Biallelic Variants Underlie a Rothmund–Thomson-like Phenotype

The two reported [2] *NUP98*-mutated adult siblings (brother and sister) presented, as above mentioned, “core” features of RTS (Table 1) [56]. WES on the five family members showed that the affected siblings inherited from their parents the *NUP98* biallelic variant c.83G > A (p.Gly28Asp) that has never been reported to date in relevant databases (Ensembl [67], GnomAD [68], 1000 genomes [69]) and leads to a replacement of the highly conserved non-polar Glycine with the charged Aspartic residue in a short linker between FG repeats of the first and largest NUP98 IDR [2]. As the mutated *NUP98* transcripts escape mRNA decay, and classic protein structural biology approaches are precluded by the dynamic structure of the FG domain, molecular modeling studies have been undertaken to characterize the mutated NUP98 FG domain compared with the wild-type (WT) [2]. This study highlighted a reduced intramolecular cohesiveness and a more elongated conformation of the mutated NUP98 FG domain compared with the WT domain (in keeping with the siblings’ senescent, but cancer-free phenotype), which is predicted to undermine its role as a multi-docking station for RNA and proteins NCT [27].

Given that FG-Nups are anchored to the NPC via linker proteins [6,16], FG-NUP98 itself acts as a linker [18] and the NUP98-RAE1 complex has a fundamental role in NTC [8], the mutated protein might not preserve the canonical “pore” functions. Furthermore, in keeping with the multitasking FG-Nups activity [70], the variable folding and dynamic behavior of the mutated NUP98 FG domain might affect “off-pore” functions including the interactions with protein complexes that function in gene transcription, chromatin remodeling [71,72], and the maintenance of genome stability [13]. It is well known that NUP98-RAE1 forms in early mitosis a complex with the multi-subunit E3 ubiquitin ligase Anaphase-Promoting Complex/Cyclosome (APC/C), bound to the Cdh1-activating subunit, and prevents unscheduled degradation of the anaphase inhibitor Securin [58]. As mentioned above, the largest C1 subunit of the APC/C molecular machine is encoded by the *ANAPC1* gene, mutated in RTS1 [57]. The complementary function of these two genes accounts for the clinical overlap of *NUP98* nucleoporopathy and RTS1, which is further strengthened by the essential role assigned to APC/C for cell proliferation and differentiation in the lens [73]. The phenotypic overlap of *NUP98*-mutated sibs with RTS and WS suggests that NUP98, ANAPC1, RECQL4, and WRN proteins are co-players in merging DNA replication and repair pathways which, when defective, lead to cell cycling alterations and genome instability [57,61,74].

The characterization of further *NUP98*-mutated individuals will permit the evaluation of the clinical similarity of *NUP98* nucleoporopathy and RTS1/RTS2/WS, and the exploration of the cellular and molecular landscape of these related disorders. Assays to monitor defective cell cycling and chromosomal/genomic instability in *NUP98* as compared with *ANAPC1*, *RECQL4*, and *WRN*-mutated cells [57,74,75] might allow for the definition of shared or unique hallmarks. The search for candidate interactors among proteins enriched in the *NUP98* mutant compared with wild-type cell lines could be the premise for enucleating the contribution of NUP98 dysfunction to these clinically overlapping syndromes.

## 7. NUP98: A Dynamic Multitasking FG-NUP

NUP98 is essential for the NCT “pore” function, making a significant contribution to the diffusion barrier facilitating NCT by mRNA export and the nuclear import/export of proteins through interactions with karyopherins and export factors [8]. Mainly positioned in the inner ring of the NPC central core, it can also be found at the cytoplasmic filaments [6,8,18] where it establishes interactions with the asymmetric CF Nups RAE1, NUP88, DDX19, GLE1, and NUP42 (Figure 2a,b) [18]. NUP98 also interacts with the nuclear basket NUP153 (Figure 2a,b), involved in post-mitotic NPC formation as assessed by the live imaging of 153-coated beads conjugated with an anti-GFP antibody which capture GFP-fused NUP98 when incorporated into telophase cells [76].

By means of its extended FG-repeats, containing 41 phenylalanine residues [77], NUP98 can simultaneously bind several nucleoporins of the inner ring and the outer rings (Figure 2a), acting as a “linker” that stabilizes the structure of the NPC [8] and contributes to the plasticity needed for the reversible constriction and dilatation of the central transport channel in response to mechanical stress on the cell [20,33]. NUP98 is one of the few symmetric Nups of human NPC whose stoichiometry is 48 instead of 32, though there is a variation of subunits across cell type, tissues, and diseases [78].

The NUP98 architecture comprises three regions. The N-ter region with the FG domain of the first and longest IDR and the GLEBS-like motif which binds the RNA export factor RAE1 (Figure 2a,c) acting in different phases of RNA export [30]. NUP98-RAE1 regulates APC/C associated with Cdh1 (Figure 2b), which in early mitosis safeguards euploidy [58] and in G2 cells controls the response to DNA damage [79]. The NUP98 middle unstructured region is characterized by motifs mediating the interactions with the inner ring Nups NUP35, NUP155, NUP188, and NUP205 of the NUP93 complex (Figure 2b,c). It has been proposed that NUP98, NUP35, and the linker region of NUP93 connect the scaffold of the four layers [20,31]. The C-ter APD (Auto Proteolytic Domain) is required to obtain the self-cleavage of NUP98 from NUP96, a component of the Y-complex of the outer rings (Figure 2a–c) [6]. To recapitulate, NUP98‘s full-length transcript is translated into a single NUP98-NUP96 precursor protein, with NUP98 and NUP96 representing the N-ter and C-ter portions, respectively [80]. NUP96 can still bind the APD after cleavage [6]. The C-ter APD is also involved in the interaction of CF NUP88 (a member of the Nup214 complex as well as NUP62) with the NUP98-RAE1 complex (Figure 2a–c). The imbalance of NUP88-NUP98-RAE1 proteins, due to NUP88 overexpression or NUP98-RAE1 haploinsufficiency, activates APC/C leading to the degradation of PLK1 kinase by NUP88 and the disruption of normal centrosome separation and aneuploidy [81]. Finally, hyper-phosphorylation of the NUP98 C-ter domain by several mitotic kinases is the decisive event for NPC disassembly and entry in mitosis [82].

How the NPC structure might be modified in *NUP98*-mutated cells, and which are the binding partners of the abnormal NUP98 protein, is actually an open question.

Like most FG-Nups, NUP98 has a propensity to self-aggregate and undergo liquid–liquid phase separation (LLPS) in vitro which can replicate the permeability properties of the NPC channel [83,84]. The essential role of FG repeats in LLPS has been supported by mutagenesis experiments and functional assays [85]. Studies on yeast have shown that the NPC FG-Nups can be categorized into two groups: those harboring FG repeats, also in the form of GLFG repeats, which are able to undergo phase separation in vitro forming liquid–liquid condensates, and FxFG Nups that do not undergo phase separation [86]. NUP98 belongs to the first group and can form co-condensates with interacting molecules and impact 3D genomic structure leading to disease. By means of this mechanism, nuclear bodies of the best characterized NUP98::HOXA9 (homeobox transcription factor) fusion protein, co-localized with the CRM1 nuclear export receptor [86], condense functional interacting molecules as the histone-methyltransferase MLL1 (also known as KMT2A) driving an aberrant chromatin structure which promotes leukemogenesis [48,49,50].

NUP98 chimeric proteins originating from genomic rearrangements are rare recurrent alterations in pediatric acute myeloid leukemia with adverse outcomes. One hundred and sixty *NUP98* rearrangements have been recorded by the comprehensive clinical and molecular characterization of 2.235 cases of AML in children/young adults (7.2%, 160 out of 2235 patients) [87].

Cellular mis-localization of FG-NUP98 to misfolded and aggregated microtubule-associated protein tau is a general feature of neurodegenerative disorders, including Alzheimer’s disease [53]. The connection between NUP98 and age-related neurodegenerative disorders has been underlined as “Nups are among the most-long living proteins in nondividing cells, such as neurons” [6].

## 8. Link of WRN/WRNIP Complex to Nucleoporins

Werner syndrome is a genome instability segmental progeria characterized by the premature onset of age-associated pathologies involving several tissues with implications in neurological dysfunction and degeneration, and an elevated risk of cancer [1,60,88]. WS is caused by biallelic null variants of the *WRN* gene encoding the WRN helicase which plays a major role in preserving genome stability functioning in DNA replication, telomere maintenance, and DNA damage repair [41]. In executing these functions, WRN is bound to several replication proteins: DNA polymerases, DNA topoisomerase I, Replication Protein A (RPA), and Proliferating Cellular Nuclear Antigen (PCNA), which associate with replication forks facilitating processivity of DNA polymerase [89]. Both helicase and exonuclease activities of WRN are essential in resolving/removing replication/repair intermediates, transcription-associated R-loops, and stalled replication forks, which occur at sites of DNA damage or G-quadruplexes (G4) formed at G-rich telomere sequences [90]. Localized within the nucleus at both nucleoli and foci of DNA replication and DNA repair, WRN is regarded as a caretaker of the genome and cells lacking WRN exhibit “variegated translocation mosaicism” as they have a propensity to develop spontaneous numerical and structural chromosomal abnormalities, including translocations, inversions, and deletions [41]. It has been proposed that the lack of WRN helicase activity results in dramatic telomere loss from individual sister chromatids, causing a DNA damage and repair response that leads to chromosome fusion-breakage cycles and genomic instability [91]. Moreover, WRN has a role in maintaining heterochromatin stability, as a global loss of histone H3 trimethylated on lys9 (H3K9me3), and changes in heterochromatin architecture have been observed in WRN-deficient cells [92].

These findings can explain WS progeroid features, supporting the concept of genome instability as a major contributor to biological aging [7,43]. WRN is structurally and functionally linked to WRNIP (Werner’s Helicase Interacting Protein), a member of the evolutionarily conserved AAA+ ATPase family [93], recognized as a factor protecting replication fork and able to promote restart after replication stress at stalled forks [94]. A classic nuclear localization signal domain represented by a short sequence of five amino acids has been mapped within WRNIP, leading to the delivery of the WRN/WRNIP complex to the nucleus [95].

Linkage of the DNA damage sensor WRNIP to the NPC has been established by its localization to the NE [64] and interaction throughout the cell cycle with the Nup107–160 sub-complex (Figure 3), known for its role in the kinetochore anchoring and in the correct bipolar spindle assembly [96].

Linkage of the WRN/WRNIP to the NPC has been confirmed by its association with the inner ring NDC1 (Nuclear Division Cycle 1) [3] (Figure 3), a universally conserved nucleoporin which anchors the inner ring to the NE by means of its transmembrane helices [97,98] and forms an interaction hub—fundamental for NPC architectural framework—with the pore POM210 (alias GP210) nucleoporin (Figure 3) [20,23] activating a DNA repair center at the nuclear periphery [99].

WRN/WRNIP double-knockout cell lines displayed an impaired DNA damage response, decreased levels of NDC1 and NDC1-interacting scaffold NUP93, and an increase, likely compensatory to maintain NPC integrity, of POM121 and ELYS, the asymmetric Y long arm CNC component localized at the nuclear face of the NPC [3] (Figure 3). These concurrent changes in NPC composition may further reduce the response at NE repair centers at the nuclear periphery [99] and impair NCT and nucleolar functional plasticity.

Indeed, in WRN/WRNIP knockout cell lines, the NE barrier is not destroyed, but the distribution of the FG-Nups and the RAN gradient, required for NCT, are changed. Processing and nucleolar localization of lamin B1, shown to interact with NDC1 and WRN [100], is also modified, leading to multiple alterations at the NPC and impacting on the functional plasticity of the nucleolus. The nucleolus, the site of WRN localization [101], is characterized by “RAN holes”, devoid of RAN and filled with condensed pools of lamin B1, usually attached to the inner NE and the neighboring cytoskeleton. Intriguingly, a breakdown of the RAN distribution has been observed in fibroblasts from Hutchinson–Gilford progeria (MIM#176670) [1], caused by mutations of the lamin A gene [102]. The link between NDC1 and the WRN complex underlines the relationship between structural changes in the NPC, nuclear compartmentalization, and genome instability.

## 9. Interactions of FAM111B Protease, Mutated in Hereditary Fibrosing Poikiloderma, and Nuclear Pore Components

Hereditary Fibrosing Poikiloderma or POIKTMP [1,65,103] is a rare autosomal dominant multisystem disorder, characterized by skin abnormalities, the hypotrichosis of hair, eyebrows, and eyelashes, tendon contractures, myopathy, lung fibrosis, and liver disease, caused by monoallelic missense variants (with a dominant negative/gain-of-function effect) of the *FAM111B* gene. *FAM111B* encodes a serine protease [65] 46% identical to the better-characterized FAM111A (Family with Sequence Similarity 111, Member A; MIM*615292 [1]) protein, which leads upon mutation to two different severe skeletal dysplasia syndromes whose molecular mechanism is unknown: perinatally lethal gracile bone dysplasia (MIM#602361) [1] and Kenny–Caffey syndrome type 2 (MIM#127000 [1]) [104,105,106,107].

The most homologous regions of FAM111B and FAM111A are the C-ter portions that harbor the catalytic trypsin-like protease domain and the two upstream ubiquitin-like domains UBL-1 and UBL-2 involved in protein–protein interactions, protein ubiquitination impacting on the proteasome system, and maintenance of cellular homeostasis [104,105]. Pathogenic *FAM111B* variants are clustered in the C-terminal catalytic region containing a D-box domain (a linear short motif or degron also identified in FAM111A) involved in proteolysis (cluster 2) and an upstream region impacting on protease activity (cluster 1). It is worth noting that a more severe phenotype, including a predisposition to pancreatic adenocarcinoma, has been associated with cluster 2 mutations [108,109].

FAM111A and FAM111B are binding partners and interact with proteins required for DNA replication such as RFC (Replication Factor C) and PCNA, although, unlike FAM111A, FAM111B does not possess a recognizable PIP (PCNA Interacting Protein box) domain and is not enriched at DNA replication sites [104]. A role in DNA replication is assigned to FAM111A as it processes protein crosslinks at DNA replication forks in response to replication stress [110]. Conversely, FAM111B is involved in preserving genome stability through functions in DNA damage repair [111] and telomere maintenance [4]. Both FAM111A and B are involved in proteostasis (dynamic regulation of waste products), whose loss is included among the primary hallmarks of cellular damage in a number of premature aging diseases [43].

POIKTMP-associated *FAM111B* alterations phenocopy the extensively characterized cellular phenotypes induced by *FAM111A* mutations, i.e., undermine cellular fitness by releasing inhibitory constraints on proteolytic activity, blocking DNA and RNA synthesis, and promoting apoptotic cell death [104,105].

As the NPC has been shown to be a FAM111A target [105], with dramatic nuclear morphology and nuclear pore distribution changes coupled with the enrichment of nucleoporin interactors (NUP153, NUP50, POM121, NUP214, NUP98, and the mRNA export cofactor GANP) in Kenny–Caffey cells with respect to wild-type [105], the same search has been applied to FAM111B. A range of FAM111B-mutated proteins localize exclusively at the nuclear periphery, instead of showing the pan-nuclear localization that characterizes the wild-type protein [4], suggesting that this location might drive the disease.

Interactions of FAM111B with nucleoporins revealed by mass spectrometry and the immunoblotting of FAM111 in overexpressing cell lines showed that the asymmetric CF FG-NUP42, functioning in transport factor binding, and SEC13, a small outer ring CNC component embedded within the unstructured N-ter extension of NUP96 (Figure 4), are likely involved in FAM111B recruitment at the nuclear periphery [4]. It is notable that the SEC13-interacting NUP96 is one of the Nups that associate with chromatin at the NPC and within the nucleoplasm [10].

Upon recruitment, FAM111B engages interactions with FG-NUP98, RAE1, and GLE1, suggesting its potential role in the RNA export machinery [4].

Overall, the FAM111B interactome is enriched in FG-Nups, including, besides NUP42 and NUP98, FG-NUP214 (Figure 4), the latter two also being interactors of FAM111A mutants [4,105]. However, none of the above interactors has yet been recognized as a substrate of the FAM111B protease, unlike FG-NUP62 which has been identified as a target of the hyperactive FAM111A protease [105]. Given that FAM111A and FAM111B are binding partners, FG-NUP62 may be indirectly involved in POIKTMP too (Figure 4). The presence of NUP98, the main regulator of APC/C in the cell cycle, in the network of FAM111B interactors, and the presence among APC/C substrates of D-box sequences [112], where most *FAM111B* (as well as *FAM111A*) pathogenic variants cluster, suggest the need for further studies to examine in more depth the connection of FAM111B with cell cycle regulation. It is worth noting that the abnormal nuclear shape (nuclear blebbing) in FAM111B defective cells recalls that observed in laminopathies [113], but the mutated FAM111B protease does not form a complex with lamins [4]. Nuclear blebbing has been associated with increased levels of endogenous DNA damage [114] though *FAM111B*-mutated cells were proficient in DNA damage response, likewise their wild type counterparts upon treatment with genotoxic compounds [4]. In spite of this, *FAM111B*-mutated cells accumulate endogenous DNA damage likely arising from replication errors and defective repair and show hallmarks of genomic instability [4]. These include abnormally short and fragile telomeres, accounted for by the scarce recruitment of the TFR2 shelterin component, suggesting that FAM111B is required for normal telomeric maintenance [4]. The critically short telomeres, often present in patients with lung fibrosis caused by primary or secondary telomeropathies [115], induce DNA synthesis in mitosis and ultra-fine DNA bridges and micronuclei in the subsequent G1 phase of the cell cycle leading to apoptosis [4]. Fragile and short telomeric ends have been characterized in Rothmund–Thomson type 2 [116] and Werner syndromes [117] linking POIKTMP to RecQ helicase defects not only via the partial clinical overlap but also through the shared hallmark of defective telomere maintenance.

## 10. Concluding Remarks

The clinical overlap between *NUP98*-mutated siblings harboring homozygous variants in the NUP98 N-ter FG domain and Rothmund–Thomson and Werner genome instability syndromes focused our attention on crosslinks between nucleoporins and genome caretakers. A review of the scope-related literature showed such connections have been reported in Werner and POIKTMP syndromes, primed by the activation of repair centers at the nuclear periphery to attenuate the vicious cycle of impaired DNA damage response and genomic instability. NPC structure and composition was found to be perturbed in WRN/WRNIP double knock-out (KO) cells compared with WT cells, while not apparently being modified in FAM111B KO compared with WT cells, though the exclusive location at the nuclear envelope of mutated FAM111B might drive the disease.

Despite the complexity of these investigations, due to the intricate network of NPC components, the limitations of the obligatory cell modeling of the syndromes under focus and the need for integrating multi-level analyses, the search for crosslinks between nucleoporins and caretakers has highlighted commonalities between two apparently discrete syndromes, which might prime new research. Both WRN and FAM111B are involved in the loading of DNA replication/repair factors, whose absence compromises DNA polymerase processivity and damage repair. Both caretakers are required for telomere elongation and it is intriguing to note that interstitial lung disease, a common sign of POIKTMP patients that correlates with short telomeres, has been developed by a patient with Werner syndrome [117].

The nucleoporins network revealed by mass spectrometry in mutated FAM111B cell lines is enriched in FG-Nups such as NUP98 and CF Nups, indicating the cooperation of FAM111B in the RNA export function. The presence of NUP98 in the POIKTMP interactome is consistent with NUP98-RAE1 regulation of the timely activity of APC/C during the cell cycle and the recognition by APC/C of the catalytic degron clustering most of the *FAM111B* mutations.

Studies on engineered *NUP98*-biallelic mutant cells could provide the clues to disclosing candidate interactors, including caretakers. In addition to the huge impact on cancer and neurodegenerative diseases of *NUP98* rearrangements which exacerbate the propensity of NUP98 N-ter to undergo phase separation, the rare *NUP98* biallelic germline mutations may shed light on accelerated aging, a process of increasing medical and societal impact.

## Figures and Tables

**Figure 1 ijms-25-09387-f001:**
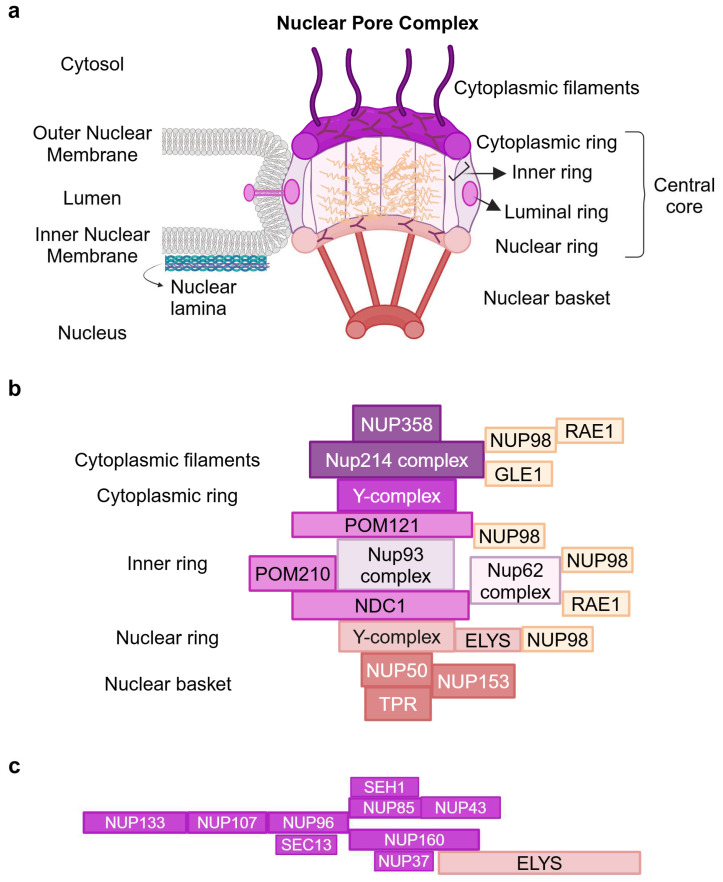
**Overview of the Nuclear Pore Complex Architecture.** (**a**) NPC central core: Cytoplasmic Ring (CR), Nuclear Ring (NR), Luminal Ring (LR), Inner Ring (IR), and the peripheral Cytoplasmic Filaments (CF) and Nuclear Basket (NB) structures protruding towards the cytosol and the nucleus. IR encircles the NPC central channel filled with the FG-Nups comprising the diffusion barrier. The rings and the peripheral structures are depicted in different colors from purple to light pink. FG repeats protruding into the channel are indicated in light orange. Adapted from “Components of the Nuclear Pore Complex”, by BioRender.com (2024). Retrieved from https://app.biorender.com/biorender-templates (30 July 2024) [21]. (**b**) Schematic of the NUP complexes of NPC central core and peripheral structures. (**c**) Schematic of the Y-shaped Coat Nucleoporin Complex (CNC) with the stem components NUP133, NUP107, NUP96, and SEC13; the short arm components SEH1, NUP85, NUP43, and the long arm components NUP37, NUP160 plus the asymmetric ELYS on the nuclear face. Figure created with BioRender.com (30 July 2024) [21].

**Figure 2 ijms-25-09387-f002:**
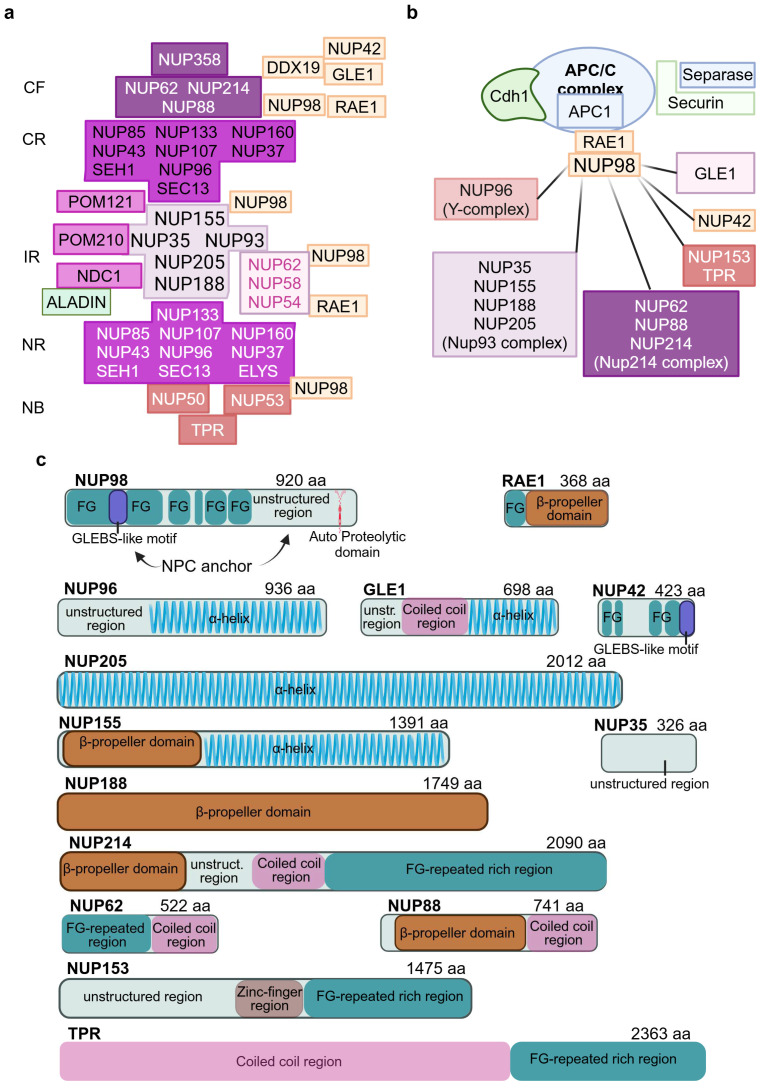
**Multiple NUP98 locations at the NPC and regulation of APC/C by NUP98-RAE1 complex.** (**a**) The dynamic NUP98 is found at multiple positions in the NPC: interacting with the CF Nups RAE1, GLE1, DDX19, NUP42, and members of the Nup214 complex; associated with the outer CR/NR Y-complex through NUP96; at the inner ring with members of the NUP93 complex and interacting with NUP153 at the nuclear basket. CF: cytoplasmic filaments, CR: central ring, IR: inner ring, NR: nuclear ring, NB: nuclear basket. (**b**) NUP98-RAE1 binds and inhibits the pre-mitotic activity of APC/C (Cyclosome/C)-Cdh1 complex. APC/C C1 subunit is encoded by *ANAPC1* gene mutated in RTS1. Other nucleoporins/Nups complexes that interact with NUP98 primarily through its C-terminal domain are depicted too. (**c**) Schematic architecture of known domains of NUP98 and interacting nucleoporins. Numbers after the nucleoporin name indicate the amino acid residues; different colors indicate the major structural motifs (FG-regions are in teal, coiled coil domain in pink, α-helical in blue, β-propeller in brown). Figure created with BioRender.com (30 July 2024) [21].

**Figure 3 ijms-25-09387-f003:**
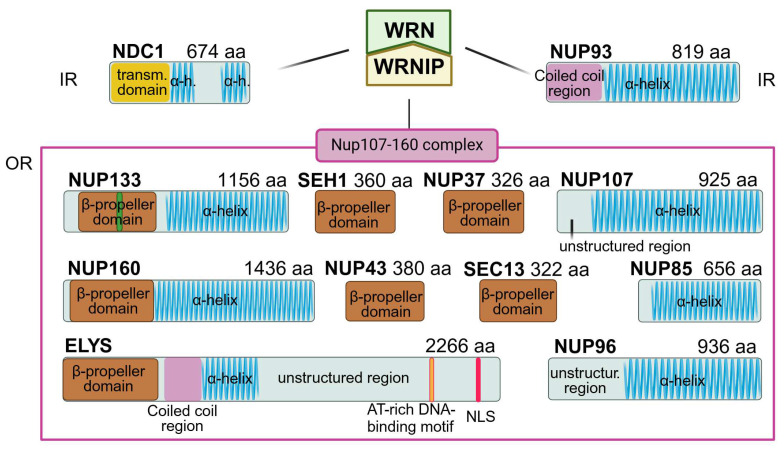
**Framework of interactions between WRN/WRNIP complex and NPC.** The WRN/WRNIP complex interacts with the Nup107–160 complex and nucleoporins NUP93 and NDC1 of the inner ring (IR). The schematic architecture of nucleoporins major known domains is shown: α-helical in blue, β-propeller in brown, coiled coil domains in pink, transmembrane domain in yellow, nuclear localization signal (NLS) in red, and AT-rich DNA-binding motif in orange. Numbers after the nucleoporin name indicate the amino acid residues. IR: inner ring, OR: outer rings. Figure created with BioRender.com (30 July 2024) [21].

**Figure 4 ijms-25-09387-f004:**
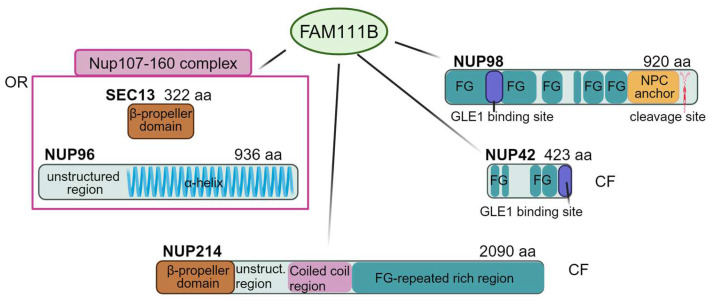
**The interconnecting network between FAM111B and NPC.** FAM111B interacts with SEC13 and NUP96 nucleoporins belonging to the Nup107–160 complex of the outer rings, the CF NUP214, NUP98, and NUP42, with the latter also located in the central channel. The schematic architecture of the major known domains of the nucleoporins is shown with FG-domains in teal, α-helical in blue, β-propeller in brown, coiled coil domains in pink, the GLEBS-like motif in indigo, and the NPC anchor in orange. Numbers after the nucleoporin name indicate the amino acid residues. OR: outer rings, CF: cytoplasmic filaments. Figure created with BioRender.com (30 July 2024) [21].

**Table 1 ijms-25-09387-t001:** Clinical features of *NUP98*-mutated siblings and Rothmund–Thomson syndrome.

Clinical Features	*NUP98* Siblings	RTS2	RTS1
Causative gene	*NUP98*	*RECQL4*	*ANAPC1*
Skin pigmentation	Mottled pigmentation since adolescence	Early poikiloderma	Early poikiloderma
Dry and fragile skin	+	+	+
Thin and fragile hair	+	+	+
Eyelashes/eyebrows	Absent	Absent	Absent
Eye abnormalities	Bilateral cataracts since infancy	−	Juvenile cataracts
Ungual abnormalities	−	+	±
Dental decay	+ since adolescence	+	±
Short stature	+ *	+	+
Skeletal abnormalities	Osteoporosis in third decade	+	+
Hypogonadism/oligomenorrhea	+	+/−	+
Malignancies	− (untill sixth decade)	Cancer predisposition Osteosarcoma/skin cancer	?

*: Only the brother; ?: not evaluated in the small cohort of characterized patients.

## Data Availability

No data were used for the research described in this article.

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
