# Peer review of "Interdependence between Nuclear Pore Gatekeepers and Genome Caretakers: Cues from Genome Instability Syndromes"

_ijms, 2024, doi:10.3390/ijms25179387_

Round 1

Reviewer 1 Report

Comments and Suggestions for Authors

This manuscript is a  review that examines the interactions between DNA damage response, genomic instability and  nuclear pore complex proteins and their associations with human disease. It is an interesting and well written review that provides a good overview of the subject together with some useful Figures describing these interactions.

It would be useful to clarify what the authors mean by “DNA damage”. This term is used repeatedly in the text but as DNA damage exists in a multitude of forms including base damage, strands breaks, and cross-links (with differing biological potencies), it would be useful to specify what the authors mean by this term and whether the meaning changes in the text. For example What is meant by “endogenous DNA damage” (line 468)? Is this the same type of DNA damage that is mentioned in lines 201 or 297?

There is an element of repetition (e.g. the repeat description of the NUP98- mutated siblings in sections 5 and 6 which should be avoided.

Author Response

Comments 1: This manuscript is a review that examines the interactions between DNA damage response, genomic instability and nuclear pore complex proteins and their associations with human disease. It is an interesting and well written review that provides a good overview of the subject together with some useful Figures describing these interactions.

Response 1: We thank the reviewer for appreciating our manuscript both in the text and the iconography.

Comments 2: It would be useful to clarify what the authors mean by “DNA damage”. This term is used repeatedly in the text but as DNA damage exists in a multitude of forms including base damage, strands breaks, and cross-links (with differing biological potencies), it would be useful to specify what the authors mean by this term and whether the meaning changes in the text. For example What is meant by “endogenous DNA damage” (line 468)? Is this the same type of DNA damage that is mentioned in lines 201 or 297?

Response 2: As the reviewer points out, we have not given punctual explanations of the DNA damage, in all forms as a basic explanation would have captured much space and would not be needed by readers interested in genomic instability syndromes. However, as correctly suggested by the reviewer, we have explained the significance of “endogenous DNA damage” (Page 16, lines 490-496) which is essential to understand the pathomechanism of POIKTMP, one of the two exemplified syndromes proving the link of nucleoporins to caretakers.

Comments 3: There is an element of repetition (e.g. the repeat description of the NUP98- mutated siblings in sections 5 and 6 which should be avoided).

Response 3: We thank the reviewer for pointing out some redundant information: we have deleted the repetition (Page 7, line 235-237) and left in Section 6 only Table 1 as clinical snapshot of the NUP98-mutated siblings.

Reviewer 2 Report

Comments and Suggestions for Authors

Major Concern: The clinical data assessed in this review is quite limited, comprising only two siblings or five family members from the same lineage. This raises the question of whether other mutations may be inherited but not clearly defined. The author should clarify the frequency of the germline biallelic variants of NUP98 (MIM*601021) in other clinical datasets to strengthen the link between nucleoporopathies (NPC) and genome instability syndromes.

Minor Concerns:

1.     Acronyms: The author should ensure that all shortened names introduced in the manuscript are clearly defined upon first use. For instance:

    • Line 17: Define WRNIP in the first place, as it is explained only until at line 338 and reiterated at line 363.
    • Line 32: Provide definitions for RECQL2 and FAM111B.

2.     Clarity in Sentences:

    • Line 53: The sentence “which form around 2000 channels in the NE of cells” is ambiguous. It may be clearer to state “around 2000 channels at the NE in one human cell.”

3.     Figure 1: In panel A, the FG repeats would be more clearly indicated in light orange to align with panel C for consistency.

4.     References: Line 89 should include PMID: 35679401 in addition to reference [21], as it pertains to the use of AI for NPC structure prediction.

5.     Figure Boundaries: In Line 91, the distinction between left and right in Figure 1b is unclear. A dashed line could be added to define these boundaries more clearly.

6.     Language Use:

    • Line 105: Replace “thanks to” with “due to” for a more formal tone.
    • Line 114: Use Å (angstrom) instead of A for measuring diameter (~200 Å).

7.     Citation Formatting:

    • Line 129: The citation “(for review see [7])” is inconsistent with its prior mention in line 52. Please standardize the formatting for reference [7].
    • Line 139: Ensure “S. cerevisiae” is italicized in the context of the discussion on DSB repair.

8.     Additional Information: In Section 4 regarding the correlation between NPC and ALS, it would enhance the discussion to include a few sentences on the role of CHMP7 (PMID: 34321318).

9.     Figure 2: Consider enlarging Figure 2 further. The labeling in Fig 2C would benefit from uniformity and enlargement for clarity. Additionally, ensure that the domain following the NPC anchor in the NUP98 architecture is clearly labeled.

Comments on the Quality of English Language

The English language can be further improved. See minor concerns above.

Author Response

Comments 1: Major Concern: The clinical data assessed in this review is quite limited, comprising only two siblings or five family members from the same lineage. This raises the question of whether other mutations may be inherited but not clearly defined. The author should clarify the frequency of the germline biallelic variants of NUP98 (MIM*601021) in other clinical datasets to strengthen the link between nucleoporopathies (NPC) and genome instability syndromes.

Response 1: We agree with the reviewer that the clinical data on germline biallelic NUP98 pathogenic variants are quite limited: 2 probands in one family characterized by segregation analysis from consanguineous carrier parents and pathogenicity according to molecular modeling used for variants affecting amino acid residues between FG repeats (Colombo 2023). The list of databases supporting the pathogenicity of the siblings’ variant reported in Colombo 2023 has been updated to 2024 ("....has never been reported to date in relevant databases (Ensembl [67], GnomAD [68], Exome Variant Server [69]" Page 7, lines 239-240). The overall genetic and phenotypic information adds NUP98 (mostly known for its somatic rearrangements in leukemia) to the growing list of nucleoporins which mutations underlie rare/ultra-rare disorders.

We have been impressed by the overlap of siblings’ phenotype to Rotmund Thomson type 1 and 2 and Werner syndrome patients. Despite the causative gene for RTS1 has been searched for almost 20 years by testing RTS RecQL4-negative patients, only 12 cases harboring biallelic ANAPC1 mutations have been characterized to date. RTS1 is an ultra-rare disease, as well as WS. We are confident that the information conveyed on NUP98 as potential candidate gene of unsolved cases of the above rare syndromes might expand the limited resources of groups working on these genome instability disorders .

Comments 2: Minor Concerns: Acronyms: The author should ensure that all shortened names introduced in the manuscript are clearly defined upon first use. For instance: Line 17: Define WRNIP in the first place, as it is explained only until at line 338 and reiterated at line 363. Line 32: Provide definitions for RECQL2 and FAM111B.

Response 2: We thank the reviewer for meticolous indications which are essential to the reader. All the reviewer suggestions have been accepted and can be visualized in the revision tracked version of the manuscript. In details, we have defined WRNIP (WRN Interacting Protein) at Page 1, line 39; WRN/RECQL2 (RecQ Protein-Like 2,) at Page 1, Line 34; FAM111B (Family with Sequence Similarity 111, Member B;..) at Page 1, Line 36; FAM111A (Family with Sequence Similarity 111, Member A; MIM*615292 [1]) at Page 15, Lines 434-435; RECQL4 (RECQ Like Helicase 4), at Page 7, Line 203.

Comments 3: Minor Concerns: Clarity in Sentences: Line 53: The sentence “which form around 2000 channels in the NE of cells” is ambiguous. It may be clearer to state “around 2000 channels at the NE in one human cell.”

Response 3: We modified the sentence as indicated (Page 2, Line 57)

Comments 4: Minor Concerns:  Figure 1: In panel A, the FG repeats would be more clearly indicated in light orange to align with panel C for consistency.

Response 4: The color of the FG repeats has been changed as suggested for consistency between the Figure panels.

Comments 5: Minor Concerns: References: Line 89 should include PMID: 35679401 in addition to reference [21], as it pertains to the use of AI for NPC structure prediction.

Response 5: We thank the reviewer for suggesting to include the reference of Fontana P when mentioning the studies employing AI for NPC structure prediction. See Page 4, Line 96 and new reference 24.

Comments 6: Minor Concerns: Figure Boundaries: In Line 91, the distinction between left and right in Figure 1b is unclear. A dashed line could be added to define these boundaries more clearly.

Response 6: We have modified Figure 1, we removed "left" and "right" from the legend and the text (Page 4, lines 88-89; Page 5, lines 98-99).

Comments 7: Minor Concerns: Language Use: Line 105: Replace “thanks to” with “due to” for a more formal tone. Line 114: Use Å (angstrom) instead of A for measuring diameter (~200 Å).

Response 7: We modified the text as suggested (Page 5, line 113; Page 5, Line 123).

Comments 8: Minor Concerns: Citation Formatting: Line 129: The citation “(for review see [7])” is inconsistent with its prior mention in line 52. Please standardize the formatting for reference [7]. Line 139: Ensure “S. cerevisiae” is italicized in the context of the discussion on DSB repair.

Response 8: We have standardized the citation formatting throughout the manuscript (Page 5, line 137; Page 6, line 162; Page 5, line 148).

Comments 9: Minor Concerns: Additional Information: In Section 4 regarding the correlation between NPC and ALS, it would enhance the discussion to include a few sentences on the role of CHMP7 (PMID: 34321318).

Response 9: We added a sentence on the enticing ALS article ("Alterations of NPC components triggered by impaired export and nuclear accumulation of CHMP7, a critical mediator of NPC, have been shown in Induced Pluripotent Stem Cell-derived ALS motoneurons, to initiate the pathological cascade leading to mis-localization of TAR DNA binding protein 43 (TDP-43), the hallmark of genetic and sporadic forms of ALS and related neurodegenerative disorders [55]." Page 6, lines 181-186, and new reference 55): this topic would deserve more space: its concise mention in a review centered on nucleoporins in genome instability syndromes underlines the wide impact of nucleoporins dysfunction on human disease.

Comments 10: Minor Concerns:  Figure 2: Consider enlarging Figure 2 further. The labeling in Fig 2C would benefit from uniformity and enlargement for clarity. Additionally, ensure that the domain following the NPC anchor in the NUP98 architecture is clearly labeled.

Response 10: We enlarged and modified Figure 2 as suggested.

Round 2

Reviewer 2 Report

Comments and Suggestions for Authors

The authors have clearly addressed my concerns.

Comments on the Quality of English Language

The quality of English language is clear.